# A Study of Nanosilver Colloid Prepared by Electrical Spark Discharge Method and Its Antifungal Control Benefits

**DOI:** 10.3390/mi12050503

**Published:** 2021-04-30

**Authors:** Kuo-Hsiung Tseng, Meng-Yun Chung, Juei-Long Chiu, Chao-Heng Tseng, Chao-Yun Liu

**Affiliations:** 1Department of Electrical Engineering, National Taipei University of Technology, Taipei 10608, Taiwan; alexmychung@gmail.com (M.-Y.C.); chiu.paul@tw.panasonic.com (J.-L.C.); 2Business Planning Development Department, Panasonic Eco Solution Sales Taiwan Co., Ltd., Taipei 10608, Taiwan; 3Institute of Environmental Engineering and Management, National Taipei University of Technology, Taipei 10608, Taiwan; tsengco@ntut.edu.tw (C.-H.T.); jasonmace_1129@yahoo.com.tw (C.-Y.L.)

**Keywords:** electrical spark discharge method, nanosilver colloid, silver ion, *Aspergillus*, yeast

## Abstract

This is a study of an antimicrobial test, including yeast, *Aspergillus Niger*, and *Aspergillus Flavus*, on a nanosilver colloid solution. The antibiosis is compared with a standard silver ion solution at the same concentration as in the experimental process. This study proved that the nanosilver colloid prepared by the electrical spark discharge method (ESDM) is free of any chemical additives, has a microbial control effect, and that the effect is much better than the Ag^+^ standard solution at the same concentration. 3M Count Plate (YM) is used to test and observe the colony counts. The microbial control test for yeast, *Aspergillus Niger*, and *Aspergillus Flavus* is implemented in the nanosilver colloid. In addition to *Aspergillus flavus*, an Ag^+^ concentration of 16 ppm is enough to inhibit the growth of the samples. At the same concentration, the nanosilver colloid has a much better microbial control effect than the Ag^+^ standard solution, which may be because the nanoparticle can release Ag^+^ continuously, so the solution using the ESDM has a more significant microbial control effect.

## 1. Introduction

Food and water are indispensable substances in human life. However, many foods become inedible or difficult to preserve due to the growth of microorganisms. Contamination by bacteria and fungi can cause a decrease in the freshness of food and cause an unpleasant taste, which can damage the food. *Aspergillus* is the main fungal contaminant and one of the important factors leading to the decay and accumulation of mycotoxins in food. The pollution of these microorganisms will not only cause huge economic damages and losses, but also pose a threat to human health. Therefore, there is an urgent need to develop safe reagents to control bacterial and fungal contamination. Due to developing technologies, nanotechnology can be one of the ways to prevent food decay. We are surrounded by the antimicrobial applications of nanosilver, in our daily necessities, textile clothing, and cosmetics, as well as in our air-conditioners and washing machines [1]. When nanosilver is used as an antimicrobial agent, it may also be hazardous to the environment and the ecology [2,3]. The FDA has warned that ingesting the nanosilver colloids may cause poisoning. In vitro research has also proven that nanosilver can harm the hepatocyte, stem cells, and brain cells of mammals [4,5,6]. Under proper concentration and control, however, nanosilver can play a great role in inhibiting microorganisms. If nanosilver is used excessively as a bacteriostatic agent, it may become hazardous to the ecological balance, as it can inhibit microbial growth at low concentrations [7,8,9].

The current so-called bactericidal mechanism of silver is that when nanosilver is close to microorganisms, it will cleave the protease that metabolizes oxygen. After losing their effectiveness, they cannot produce normal metabolism for oxygen, leading to their natural death, and finally they will be excreted from the body with the body’s metabolism. This is different from general bactericidal drugs, which also kill the good bacteria in the body. After nanosilver leaves the body, the cells remain intact, so nanosilver is safe for humans, reptiles, plants, and other multicellular organisms.

This study used the electrical spark discharge method (ESDM) to prepare nanosilver colloids and obtained good research findings. As the nanosilver colloid that is produced can release silver ions [10,11,12], and the nanosilver colloid is prepared by using electric sparks, it is free of any chemical substances, and it is relatively friendly towards the environment and the human body. It can be used directly in medical treatments in the future; for example, it can be used in organisms or made into dressings that have a microbial control agent. This study aimed to examine the effects of the nanosilver colloid prepared by the electrical spark discharge method on yeast, *Aspergillus niger*, and *Aspergillus flavus* resistance.

## 2. Materials and Methods

### 2.1. Preparation of Nanosilver Colloid by Using the Electrical Spark Discharge Method

The architecture of the nanometer metallic colloid preparation system is shown in Figure 1. The upper and lower silver electrodes are immersed in deionized water (DW) [13,14]. After setting the servo control system parameters and establishing the architecture, the spark discharge can be initiated. The surfaces of the electrodes are melted at a high temperature in order to generate nanometer metallic particles, which are collected by the DW to create the nanosilver colloid [15]. The nanometer particles can be stably suspended long term in DW, due to the static force equilibrium [16]. There will also be a beam under the radiation of the laser light, known as the Tyndall effect [17,18].

The ESDM system can use an oscillator to observe the voltage and current wavelengths between the electrodes, as shown in Figure 2. Where T_on_ is the time of conduction when the electrodes will successfully carry out arc discharge, T_off_ is the time of stop, and after a successful electrode discharge, the DW should be restored to the state of insulation in order to facilitate the discharge of the next cycle and to eliminate the metallic particles [19].

The parameter settings of the ESDM are described as follows. The 99.99% pure Ag was selected as electrodes. The two electrodes were cleaned with DW and fixed in the ESDM and beaker, and the upper and lower electrodes were aligned. T_ON_-T_OFF_ was adjusted to 50–50 µs, and preparation time was performed for 40 min. The electrodes were weight before and after the preparation, and the results are shown in Table 1. After that, the prepared nanosilver colloid was kept still for 2–3 weeks and filtered through filter paper, during which about 77.28 mg of precipitates and big silver particles were removed. The estimated concentration was (124.43 − 77.28)/0.25 = 308 ppm.

### 2.2. Experiment Methods and Design

The experimental process was approximately divided into two parts; one was the preparation of the nanosilver colloid and the strain activation and cultivation, while the other was the microbial control cross testing of the prepared test solution and strain. It is described in the following four steps:Preparation of testing solutions: the nanosilver colloid was prepared by ESDM (No. 1 test solution, which contained 32 ppm Ag^+^ and 300 ppm nanoparticle), as samples for microbial control testing. No. 2 was a 32 ppm Ag^+^ standard solution.Activation and cultivation of strains: there were three kinds of test subject, including *Aspergillus niger* and *Aspergillus flavus* bought from BRBC, and yeast for making steamed bread. As *Aspergillus niger* and *Aspergillus flavus* were stored in dry glass tubes for a long period, secondary offspring cultivation had to be implemented for the species, in order to guarantee the strain quality.Preparation of a spore suspension: 10 μL inoculating loop (or transferring loop, loop for short) was used, one loop of fungi was scraped and put in 25 mL, 0.05% Tween80 solution to make the test solution, and the concentration was adjusted to 10^−2^ dilutions and 10^−4^ dilutions, for future use.3M Petrifilm™ Yeast and Mold Count Plate (petrifilm) was used for microbial control testing [20,21]: the test solution was added in the prepared spore suspension, the test solution concentration was adjusted to 32 ppm, 16 ppm, 8 ppm, 4 ppm, and 2 ppm, where a 1 mL test solution was drawn by pipette and dripped onto the petrifilm, and placed in an incubator for observation.

### 2.3. Preparation and Characteristic Data of the Nanosilver Colloid

The No. 1 solution used ESDM to prepare the nanosilver colloid, and the related parameter settings and characteristics are described. UV–Visible Spectroscopy (Thermo Spectronic, Helios Alpha 9423 UVA 1002E, Waltham, MA, USA) uses visible light and UV tubes as a light source. Nanoparticles have surface plasmon resonance (SPR) in light [22]. Different metal particles have particular and strong light absorption characteristics, which are influenced by the surface adsorption molecules. Therefore, during the experimentation, the nanoparticles can be analyzed by a spectrometer in order to identify the type and characteristics of the nanoparticles. The UV absorption wavelength of the nanosilver colloid prepared by ESDM is shown in Figure 3a. The absorption was around 1.13 at the wavelength of 390 nm.

The laser light scatterometer (Malvern Instruments Ltd., Nano-ZS90, Worcestershire, UK) uses non-invasive back scattering (NIBS) to detect a scattered light source [23,24]. The laser light scatterometer can measure the surface potential of particles, where the measuring principle is electrophoresis. When an external voltage is applied to the nanofluid, the charged nanoparticles display the panting phenomenon in the fluid. When the applied voltage reaches equilibrium with the viscous force of the fluid, the fluid panting velocity can be measured, and the surface potential value of particles is obtained. Zeta potential is a distribution based on intensity. It indicates how much light (or scattered intensity or photon count) contributes to the signal from various dominant zeta potentials. The zeta potential wavelength of the nanosilver colloid prepared by ESDM is shown in Figure 3b. The average zeta potential was −22.3 mV ± 4.77 mV. Figure 3c presents a field-emission scanning electron microscopy (FE-SEM) image of the prepared nanosilver colloid [25]. The colloidal particles were almost spherical, and the particle size was approximately 30 nm in diameter. The energy was as high as 15.0 kV. The model of FE-SEM was SEM HITACHI Regulus 8100 FE-SEM.

### 2.4. Activation and Cultivation of Strains

There were three sample types: yeast, *Aspergillus niger*, and *Aspergillus flavus*. The *Aspergillus niger* and *Aspergillus flavus* were pure strains bought from BCRC (Bioresource Collection and Research Center) [26], while the yeast, which was activated by 42 °C hot water and a little sugar for 6 h, was the universal brand that is available on the market (Globe Horse, USA).

The activation procedures of *Aspergillus niger* and *Aspergillus flavus* are described as follows:Cultivation of the first-generation species: 0.3–0.5 mL specified liquid medium was dripped into the unsealed *Aspergillus niger* and *Aspergillus flavus* drying tubes. The inner tube *Aspergillus* were washed off, and the 0.1 mL bacteria solution suspension was instilled in the center of solid medium. The culture medium for *Aspergillus niger* was malt extract agar (MEA), while that for *Aspergillus flavus* was potato dextrose agar (PDA), which were uniformly smeared by an L-shaped glass rod, placed in an incubator at the ambient temperature of 25 °C, and cultured for 7–9 days.Cultivation of the second-generation species: The cultivation of the second-generation species is also known as secondary offspring cultivation, where uncontaminated and complete colonies were selected from the culture medium of the first-generation species, about 0.5 cm^2^ was scraped and uniformly smeared by an L-shaped glass rod, and cultivated in the incubator until the mold spores grew. The experiment was conducted on a sterile console.

### 2.5. Experimental Design and Procedure

Petrifilm was used to test and observe the colony counts, and it involved the following:The loop was used to scrape the cultured strain off the culture dish, it was soaked in a 0.05%, 25 mL Tween 80 solution, and the solution was diluted 100× and 10,000×, respectively.A pipette was used to extract 0.1 mL of the solution diluted 100×, which was placed under a digital microscope. The concentration was observed and adjusted with DI water, and the number of strains was smaller than 600 in order to maintain the optimum observed number of strains on the count plate below 300.The No. 1 and No. 2 test solutions were mixed with quantitative bacteria solution to prepare the required concentration.The pipette was used to extract 1 mL of the prepared solution, which was injected onto the count plate, covered with the membrane, flattened by a press plate, placed in the incubator, and observed regularly.

## 3. Results

### Research on the Effectiveness of the Microbial Control of Nanosilver Colloid

The symbol explanation is shown in Table 2. The residual amount of the number of colonies is defined as Equation (1).
(1)$$\mathrm{RP}=\frac{R}{T}\times 100\%$$
whereRP: residual amount of the number of colonies by percentage;T: total amount of colonies in 2 ppm concentration; andR: residual amount of the number of colonies.

The concentrations of the nanosilver colloid and ion silver solutions were tested using a count plate, and the test results are described in Figure 4. The colony count of *Aspergillus niger* is shown in Table 3 and Figure 5, the colony count of *Aspergillus flavus* is shown in Table 4 and Figure 6, and the colony count of yeast is shown in Table 5 and Figure 7.

By using the ESDM, the nanosilver colloid could be prepared. This method is costless, simple, and free of chemical agents. In this study, the method applied in synthesizing nanosilver colloid allowed antimicrobial activity, including yeast, *Aspergillus niger*, and *Aspergillus flavus*. Using the 3M Count Plate can easily achieve the counting results. In the *Aspergillus niger* test, the nanosilver colloid had more of an effect even though there was some deviation of the test. The 16 ppm nAg and Ag showed no *A. niger* on the colony count plate. In the *Aspergillus flavus* test, under the same concentration, the nanosilver colloid had a better microbial control effect. Even with 32 ppm colloid and Ag^+^, the *A. flavus* still grew but a concentration of 16 ppm can inhibit the growth. In the yeast test, 8 ppm of the nanosilver colloid prevented the yeast from growing.

## 4. Discussion

*Aspergillus* is a ubiquitous fungus that grows on stored grains. Toxins produced by some species can harm human and animal health and cause liver and kidney toxicity, immunosuppression, and carcinogenicity. The main fungicides used to prevent the growth of fungi may be toxic to humans and their repeated use will increase the resistance of microorganisms over time. Nanosilver has antimicrobial results, including yeast, *Aspergillus niger*, and *Aspergillus flavus*. By using the 3M Count Plate, the results were easily achieved. The results of the microbial control test by count plate are described as follows (see Figure 8):Compared with the number of colonies with a 2 ppm concentration, the higher the concentration, the lower the amount of residual colonies.With addition to *Aspergillus flavus*, the nanosilver colloid had a better microbial control effect than Ag^+^, at the same concentration.With addition to *Aspergillus flavus*, all strains had a 32 ppm concentration. The residual RP was less than 0.1%.Effect of the microbial control of nano silver and silver ions, in order, was: yeast > *Aspergillus niger* > *Aspergillus flavus.*

## 5. Conclusions

A microbial control test was conducted with nanosilver colloids on *Aspergillus niger*, *Aspergillus flavus*, and yeast. The results are summarized as follows:The nanosilver colloid was prepared using the electrical spark discharge method, which is free of any chemical additives. The absorption was around 1.13 at the wavelength of 390 nm and the average zeta potential was −22.3 mV ± 4.77 mV. With the result of the SEM, the particle size was approximately 30 nm in diameter.The nanosilver colloid prepared by ESDM had a microbial control effect, and the effect was much better than the Ag^+^ standard solution, at the same concentration.Among all the test samples, the Ag^+^ concentration of 16 ppm was enough to inhibit the growth of *Aspergillus niger*, *Aspergillus flavus*, and yeast. The antimicrobial effect of nanosilver and Ag^+^ was, in order of strength, yeast > *Aspergillus niger* > *Aspergillus flavus*.At the same concentration, the nanosilver colloid had a much better microbial control effect than the Ag^+^ standard solution because the nanoparticle can release Ag^+^ continuously, such that the No. 1 test solution had a more significant microbial control effect.

## Figures and Tables

**Figure 1 micromachines-12-00503-f001:**
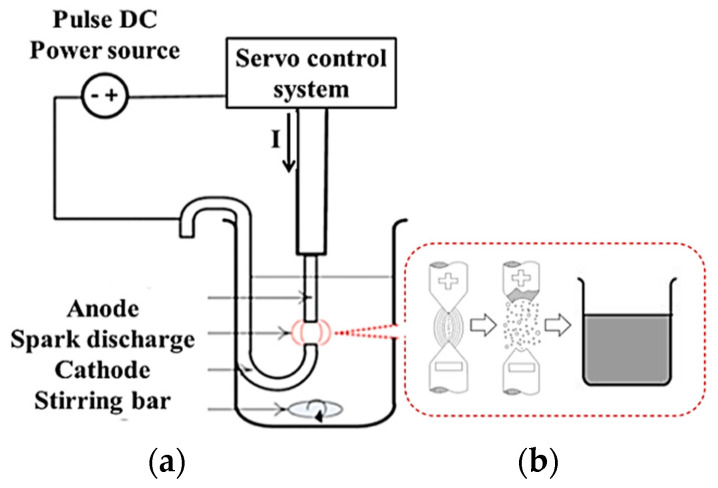
System of the electrical spark discharge method: (**a**) nanometer metallic fluid preparation system and (**b**) picture of the ESDM system.

**Figure 2 micromachines-12-00503-f002:**
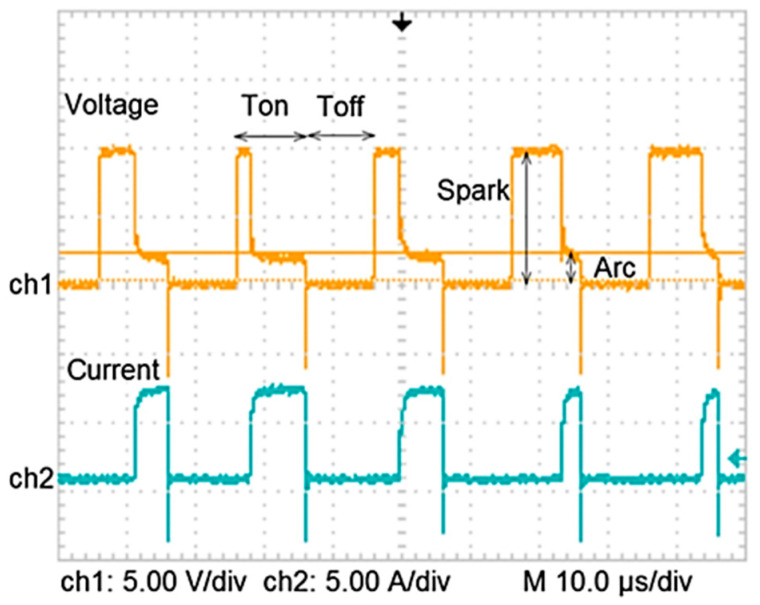
Diagram of the Ton-Toff 50-50µs discharge waveform.

**Figure 3 micromachines-12-00503-f003:**
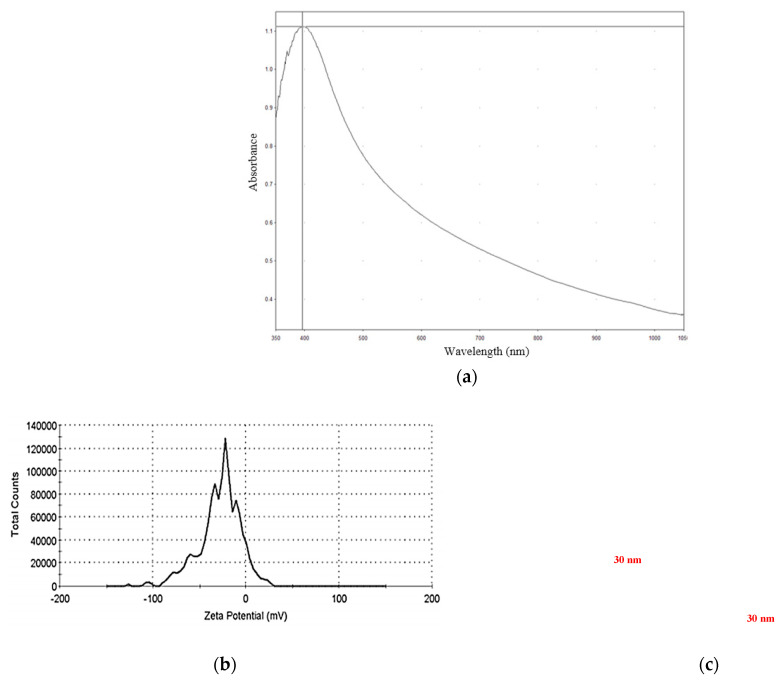
Nanosilver colloid prepared by ESDM: (**a**) UV absorption wavelength; (**b**) zeta potential distribution; and (**c**) FE-SEM image.

**Figure 4 micromachines-12-00503-f004:**
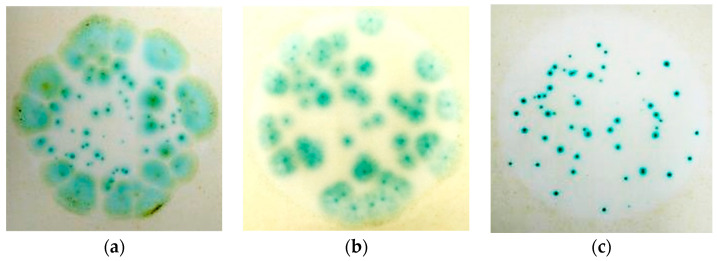
Different fungus reactions in the 3M Count Plate: (**a**) *Aspergillus niger*, (**b**) *Aspergillus flavus*, and (**c**) yeast.

**Figure 5 micromachines-12-00503-f005:**
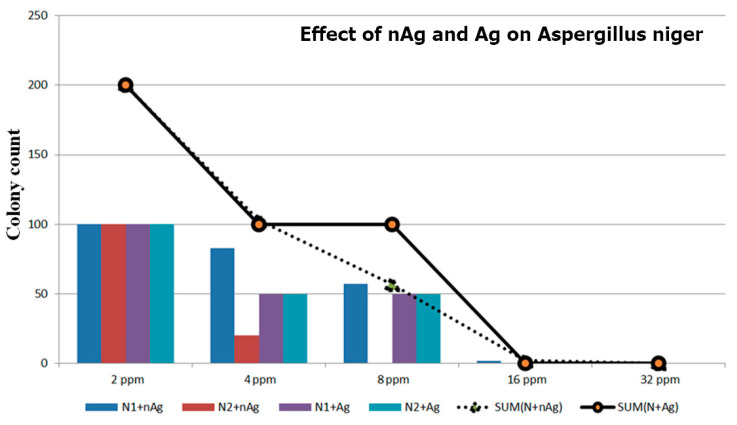
Effect of nAg and Ag on *Aspergillus niger*.

**Figure 6 micromachines-12-00503-f006:**
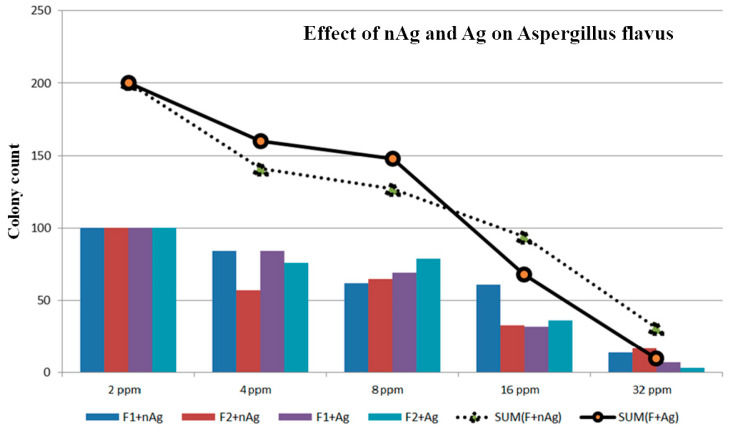
Effect of nAg and Ag on *Aspergillus flavus*.

**Figure 7 micromachines-12-00503-f007:**
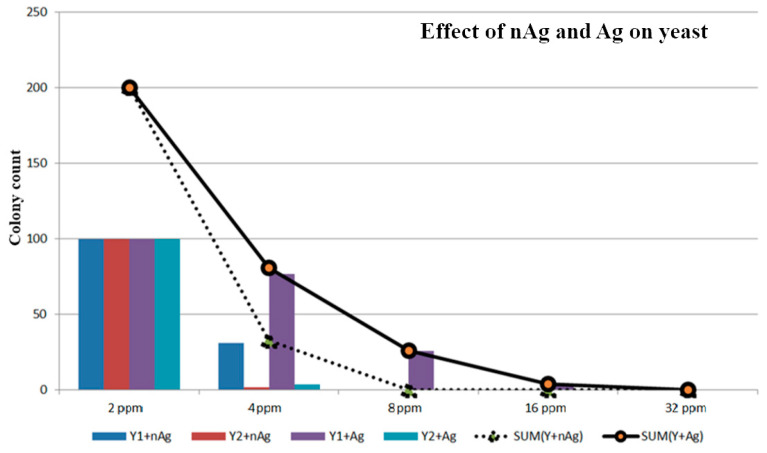
Effect of nAg and Ag on yeast.

**Figure 8 micromachines-12-00503-f008:**
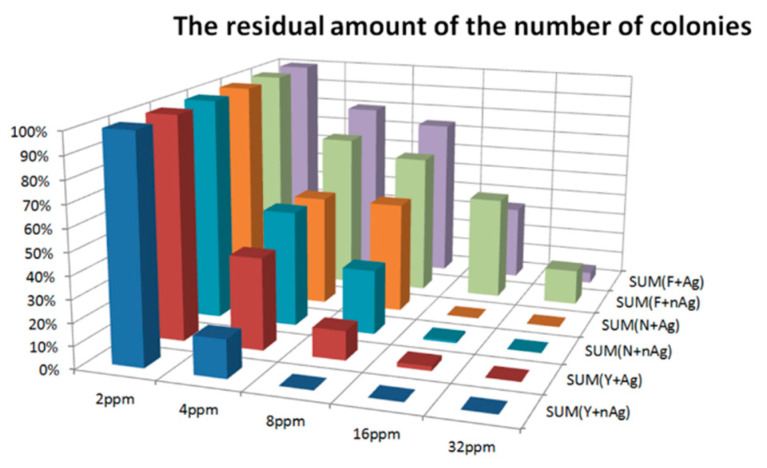
The residual amount of the number of colonies (%).

**Table 1 micromachines-12-00503-t001:** Weight loss calculation of the electrode.

Ag (Mg)	Before (W_0_)	After (W_1_)	W_0_ − W_1_ (Mg)
Electrode E_1_ (anode)	640.70	518.49	122.21
Electrode E_2_ (cathode)	5775.78	5773.56	2.22
E_1_ + E_2_	6416.48	6292.05	124.43

**Table 2 micromachines-12-00503-t002:** Symbol explanation.

①	②		③	④			
N	1	+	nAg	2	①	N	*Aspergillus niger*
						F	*Aspergillus flavus*
				4		Y	Yeast
F				8	②	1	10^−2^ dilution
						2	10^−4^ dilution
Y	2		Ag	16	③	nAg	nanosilver colloid
				32		Ag	ion silver solution
					④	2–32	concentration (by ppm)

**Table 3 micromachines-12-00503-t003:** Colony count of *Aspergillus niger*.

	2 ppm		4 ppm		8 ppm		16 ppm		32 ppm	
	No.	RP	No.	RP	No.	RP	No.	RP	No.	RP
**N1 + nAg**	120	100	100	83	68	57	2	2	0	0
**N2 + nAg**	20	100	4	20	0	0	0	0	0	0
**SUM(N + nAg)**	---	200	---	103	---	57	---	2	---	0
**N1 + Ag**	2	100	1	50	1	50	0	0	0	0
**N2 + Ag**	2	100	1	50	1	50	0	0	0	0
**SUM(N + Ag)**	---	200	---	100	---	100	---	0	---	0

**Table 4 micromachines-12-00503-t004:** Colony count of *Aspergillus flavus*.

	2 ppm		4 ppm		8 ppm		16 ppm		32 ppm	
	No.	RP	No.	RP	No.	RP	No.	RP	No.	RP
**F1 + nAg**	340	100	285	84	212	62	206	61	47	14
**F2 + nAg**	72	100	41	57	47	65	24	33	12	17
**SUM(F + nAg)**	---	200	---	141	---	127	---	94	---	31
**F1 + Ag**	340	100	285	84	236	69	108	32	24	7
**F2 + Ag**	33	100	25	76	26	79	12	36	1	3
**SUM(F + Ag)**	---	200	---	160	---	148	---	68	---	10

**Table 5 micromachines-12-00503-t005:** Colony count of yeast.

	2 ppm		4 ppm		8 ppm		16 ppm		32 ppm	
	No.	RP	No.	RP	No.	RP	No.	RP	No.	RP
**Y1 + nAg**	1520	100	475	31	2	0	0	0	0	0
**Y2 + nAg**	55	100	1	2	0	0	0	0	0	0
**SUM(Y + nAg)**	---	200	---	33	---	0	---	0	---	0
**Y1 + Ag**	1520	100	1176	77	402	26	54	4	3	0
**Y2 + Ag**	1017	100	41	4	4	0	0	0	0	0
**SUM(Y + Ag)**	---	200	---	81	---	26	---	4	---	0

## Data Availability

No new data were created or analyzed in this study. Data sharing is not applicable to this article.

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
