# Peer review of "A Study of Nanosilver Colloid Prepared by Electrical Spark Discharge Method and Its Antifungal Control Benefits"

_micromachines, 2021, doi:10.3390/mi12050503_

Round 1

Reviewer 1 Report

The title and abstract state that the article describes the antimicrobial property of the colloidal solution of silver nanoparticles. The introduction consists of a state of the art describing only the antibacterial activity of silver nanoparticles. Antimicrobial and antibacterial are not the same thing. Antimicrobial refers to all types of microorganisms while antibacterial refers only to bacteria. The introduction should also contain information about the antifungal activity of silver nanoparticles.

                At the same time the testes are only done on fungi, yeast and two different species of moulds from the Aspergillus genus, which are eukaryotic cells with different structures compared to bacteria. The antimicrobial tests should also contain at least some antibacterial tests, or the title should be changed to antifungal activity.

                The results need to be described more clearly. These include a physico-chemical and a biological characterization. I also believe that the results need to be improved with additional tests, at least TEM / SEM, for silver nanoparticles.

                The correlation of conclusions and results with the information in the introduction should be specified more clearly.

                It is not understood how the concentration of the colloidal solution of silver nanoparticles obtained by ESDM was determined.

Author Response

Please see the attachment for detail reply.

Reviewer 2 Report

The introduction needs to be improved. Discuss more about the current state and what is new about this paper.

Add more current references 2020 - 2021 regarding the approached field

Figures 4, 5, 6, 7 are not very unclear, the writing is very small. Either improve the quality or replace them.

In each subchapter, try to detail more. In a subchapter, end with a sentence, don't let it end in a figure or text.

Results and discussions need to be improved.

Author Response

(The authors gave the same response as above.)

Reviewer 3 Report

Although the topic may be interesting in some aspects, the reviewer consider that more research and investigation is needed to come to such conclusions. In fact, other techniques (i.e.  some more sophisticated) could give an additional value to the  study and going more in depth in the analysis of inhibiting the growth of the microbial samples.

For this reason the reviewer rejects this article for publishing in such state and suggests a drastic improvement of the goal of the paper.

A summary of the weak points of the article is written below:

  • When you say: “This study proved that the nanosil-11 ver colloid prepared by Electrical Spark Discharge Method (ESDM) has microbial control effect, and 12 the effect is much better than the Ag+ standard solution at the same concentration”                                                                It is clear that you are explaining the aims achieved in the experiment. However, saying that with this study:

“The nanosilver colloid prepared by Electrical Spark Discharge Method which is free of any chemical additive. And in comparison, to other nano-preparation methods,  it is more applicable to biotechnology, even to the human body.

You are making some mistakes: you cannot extrapolate that the The nanosilver colloid prepared by Electrical Spark Discharge Method is more applicable to biotechnology. There are other factors that nanosilver ions may affect seriously to the normal operation of an organism such as the human being (radicals may produce cancer, alter the ionic concentration of the organism, etc.)

  • There are many errors in describing the experiment. English writing should be improved: you cannot write an article as if it were a proof of concept. Description of the results should be much better performed instead of enumerating the ideas.
  • In Figure 3b, the units of the Y axis are not clear what they are.
  • The organization of the article is not clear. The table 1 caption should be better positioned in the manuscript. What are the Y axis units in Figure  5 and 7?
  • English writing should be improved.

Author Response

(The authors gave the same response as above.)

Round 2

Reviewer 1 Report

This version is much improved. The authors took into account all the recommendations regarding the first version of the paper.

Author Response

Dear Reviewer,

Thank you for all the help and the useful comments. We are really appreciated it.

Sincerely,

Dr. Kuo-Hsiung Tseng

Professor,

Department of Electrical Engineering,

National Taipei University of Technology

Reviewer 2 Report

Article was well improved.

Author Response

(The authors gave the same response as above.)

Reviewer 3 Report

The article style has improved. English writing too. 

However i do no understand the following:

"This study used Electrical Spark Discharge Method (ESDM)electric sparks to preparenanosilver colloids, and has obtained good research findings. As the produced nanosilver colloid that is produced can release silver ions [109-1211], and the nanosilver colloid is prepared by using electric sparks, it is free of any chemical substances, and it is relatively friendly towards the environment and the human body. It can be directly used directly in
medical treatments in the future, ; for example, it can be used in organisms, or made into a dressings with that have a microbial control agent. "

How do you prove that this nanosilver colloids  is relatively friendly towards the environment and the human body? I do not see any experiment in this publication  where the organism does not suffer any reaction or rejection when the coloids are in the human/animal body.

Author Response

Thank you for the comment. Please see the attachment.
